# Pharmacogenetics of Long-Term Outcomes of Schizophrenia Spectrum Disorders: The Functional Role of CYP2D6 and CYP2C19

**DOI:** 10.3390/jpm13091354

**Published:** 2023-09-04

**Authors:** Amrit K. Sandhu, Elnaz Naderi, Morenika J. Wijninga, Edith J. Liemburg, Danielle Cath, Richard Bruggeman, Behrooz Z. Alizadeh

**Affiliations:** 1Department of Epidemiology, University Medical Centre Groningen, University of Groningen, 9713 GZ Groningen, The Netherlands; 2Centre for Statistical Genetics, Gertude H. Sergiesky Centre, Department of Neurology, Columbia University Medical Centre, New York, NY 10032, USA; 3Department of Psychiatry, University Medical Center Groningen, University of Groningen, 9713 GZ Groningen, The Netherlands; 4GGZ Drenthe, Department of Specialist Trainings, 9704 LA Assen, The Netherlands

**Keywords:** schizophrenia, pharmacogenetic, polygenic risk score, antipsychotic, pharmacokinetic, pharmacodynamic, metabolism, cardiometabolic syndrome, genotyping, Cytochrome P450

## Abstract

Schizophrenia spectrum disorders (SSD) are complex mental disorders, and while treatment with antipsychotics is important, many patients do not respond or develop serious side effects. Genetic variation has been shown to play a considerable role in determining an individual’s response to antipsychotic medication. However, previous pharmacogenetic (PGx) studies have been limited by small sample sizes, lack of consensus regarding relevant genetic variants, and cross-sectional designs. The current study aimed to investigate the association between PGx variants and long-term clinical outcomes in 691 patients of European ancestry with SSD. Using evidence from the literature on candidate genes involved in antipsychotic pharmacodynamics, we created a polygenic risk score (PRS) to investigate its association with clinical outcomes. We also created PRS using core variants of psychotropic drug metabolism enzymes CYP2D6 and CYP2C19. Furthermore, the CYP2D6 and CYP2C19 functional activity scores were calculated to determine the relationship between metabolism and clinical outcomes. We found no association for PGx PRSs and clinical outcomes; however, an association was found with CYP2D6 activity scores by the traditional method. Higher CYP2D6 metabolism was associated with high positive and high cognitive impairment groups relative to low symptom severity groups. These findings highlight the need to test PGx efficacy with different symptom domains. More evidence is needed before pharmacogenetic variation can contribute to personalized treatment plans.

## 1. Introduction

Schizophrenia spectrum disorders (SSD) are severe mental disorders that affect around 1% of the global population [1]. Various combinations of positive symptoms (i.e., hallucinations), negative symptoms (i.e., social withdrawal), and cognitive impairments (e.g., memory problems) contribute to its impairments on quality of life and mortality [1,2]. Despite antipsychotic medication being the mainstay treatment, their effectiveness is highly variable; up to 50% of patients do not experience symptom relief or experience adverse drug effects [3,4,5]. The first generation of typical antipsychotics (TAP), including amongst others chlorpromazine and haloperidol, is effective in improving positive symptoms. However, TAP often causes motor or extrapyramidal side effects (i.e., tardive dyskinesia) [3,6]. Atypical antipsychotics (AAP), such as olanzapine and risperidone, may cause cardiometabolic complications [6,7]. Clozapine, another AAP, is licensed for treatment-resistant SSD and requires monitoring for blood abnormalities as it may cause agranulocytosis [6]. Both classes of antipsychotics have shown limited efficacy for negative symptoms and cognitive impairments, which could be attributed to the existence of distinct pathophysiological mechanisms [7,8,9]. The failure of antipsychotic treatment has been attributed to clinical (e.g., age of onset), demographic (e.g., gender), environmental (e.g., smoking), and genetic factors [4,5]. 

Pharmacogenetics (PGx) aims at optimizing the application of antipsychotics in SSD by tailoring treatment to individuals’ genetic profiles [10,11]. Specifically, PGx investigates how gene variants affect drug absorption, metabolism, and action to optimize treatment by identifying biomarkers that can predict medication response and tolerability [10,11]. Traditional PGx research has focused on single nucleotide variants (SNVs) involved in the pharmacodynamics of antipsychotics, such as those affecting dopamine (DA) and serotonin (5-HT) receptor systems, but these have shown limited clinical utility due to their weak effect sizes [3,12]. More recently, polygenic risk scores (PRS) have become a powerful tool to integrate multiple risk SNVs that may predict treatment response [13].

However, PGx PRS studies in antipsychotic response have been limited over the past two decades. In 2000, Arnaz and colleagues identified six serotonin receptor related SNVs that predicted clozapine response in SSD patients with a 76.7% success rate [14]. However, this approach did not systematically incorporate all SNVs. In 2016, Zhang et al. performed a meta-analysis and found a significant association between a PRS of six SNVs and antipsychotic-induced weight gain (AIWG) [15]. Due to the extensive meta-analysis required to consolidate reliable findings from individual candidate gene studies, this method is frequently underutilized. As a result, there is a growing trend of using PRS for disease-specific traits, such as SSD PRS, to predict treatment response [16]. This approach might capture a small true effect which may be confounded by the disease SNVs themselves. Furthermore, these PRSs often overlook important PGx SNVs involved in drug metabolism, such as those of CYP2D6 and CYP2C19 enzymes [16,17,18,19,20]. 

Rare functional SNVs in these enzymes account for a significant proportion of genetic variability in antipsychotic metabolism and can be used to classify an individual’s metabolizing phenotype (i.e., genotype–phenotype prediction) as poor (PM), intermediate (IM), normal/extensive (NM), and rapid/ultra-rapid (RM/UM) [21]. Individuals with CYP2D6 PM, which occurs in 10% of Europeans, are thought to be at significantly higher risk for adverse drug events and lack of response to antipsychotics compared to other metabolizers [12]. Consequently, drug regulatory agencies have developed dosing guidelines for these genes, although there is poor agreement amongst them [22]. For example, the Dutch Pharmacogenetics Working Group has provided dosing guidelines for CYP2D6 metabolism and several antipsychotics, but only two are deemed actionable by the US Food and Drug Administration and the European Medicines Agency [22]. The lack of consensus amongst regulatory bodies may attributed to inconsistent research findings on the true effect of these genetic variations [22].

The current study sought to investigate the effects of PGx genes PRS on long-term clinical outcomes in SSD, including cognitive, positive, and negative symptom trajectories, as well as on cardiometabolic complications. Our approach involved creating PRS based on key PGx SNVs, such as those found in genes responsible for antipsychotic response and metabolism. Additionally, we delved into the functional antipsychotic metabolizing status of patients, specifically their CYP genotype predicted activity scores, and analyzed its impact on the clinical outcomes. 

We hypothesized that combining significant SNVs identified from a meta-analysis of candidate gene studies into a PRS would increase predictive power and result in a significant association with clinical outcomes. Furthermore, we proposed that PRS for CYP2D6 and CYP2C19 enzymes would be effective in determining an individual’s metabolizer status and would be associated with clinical outcomes. This study took on a polygenic approach to determine the involvement of key PGx variants on long-term clinical symptom trajectories and cardiometabolic parameters. 

## 2. Materials and Methods

### 2.1. Study Design and Participants 

The present study is situated within the framework of the genetic risk and outcome of psychosis (GROUP) cohort [23]. GROUP is a comprehensive, six-year, prospective national study in the Netherlands, which aimed to investigate the roles of genetic and environmental factors that contribute to the expression of psychosis. GROUP involved 36 partner mental institutions and university medical centers (Amsterdam, Groningen, Utrecht, Maastricht). Clinicians working within these institutions screened patients for eligibility based on the following inclusion criteria: (1) Have schizophrenia spectrum disorder according to the 4th edition of the Diagnostic and Statistical Manual of Mental Disorders (DSM-IV), (2) be between the ages of 16–50, (3) be Dutch speakers, and (4) be able to provide informed consent [2]. In addition to patients, unaffected first-degree relatives of patients and unrelated healthy controls were also sampled. Clinical outcome data were collected at one of the respective institutions or medical centers at baseline (T1), three (T2), and six (T3) years, along with genotype data. Further details on the GROUP cohort can be found in a previous publication [23].

### 2.2. Outcome Measures 

To assess the severity of positive and negative symptoms in patients, the widely-used valid positive and negative syndrome scale (PANSS) for Schizophrenia was employed [24]. The PANSS entails a clinician-administered interview comprising 30 items, with responses measured on a seven-point Likert scale [24]. Two subscales were derived from the PANSS based on the European Psychiatric Association guidelines [25]. Positive symptom items were summed to create the positive subscale (P1–P7) and core negative symptom items (N1-4, N6) were summed to create the negative subscale. To measure neurocognitive function, a composite score was generated by averaging the Z-score of the following tests: World learning task (WLT) (immediate recall, delayed recall), continuous performance test-HQ (CPT-HQ) (CPT performance index and CPT variability), and the Wechsler adult intelligence scale (WAIS-II) (digit symbol substitution, information, arithmetic, and block design) [26,27,28]. Group-based trajectory modeling was used to identify distinct subgroups with unique symptom trajectories over time (up to 6-years follow-up) [29]. To increase statistical power for the current analysis, the symptom clusters were merged. Positive and negative symptom trajectories were dichotomized into high and low symptom groups, while cognition impairments were consolidated into high, mild, and no impairment groups. This resulted in sum scores <15 for both positive and negative symptoms classifying as low severity and >15 were for high symptom severity. For the cognitive composite scores, high impairments were <−2, mild impairments were 1 to −1, and no impairments were >1.5. In addition, we collected cardiometabolic variables at T2 (3-year follow-up), including glycated hemoglobin levels (HbA1c) in mmol/mol, body mass index (BMI) in kg/m^2^, umbilical waist circumference (cm), blood pressure (mmHg), pulse rate (beats/min), triglycerides (mmol/L), high-density lipoprotein (HDL) in mmol/L, and low-density lipoprotein (LDL) in mmol/L.

### 2.3. Covariables

Antipsychotic medication use was obtained and classified as either atypical (AAP) or typical antipsychotics (TAP) due to no single antipsychotic meeting the minimum sample size requirements [30]. The medication use was categorized into three groups: AAP if this was recorded at all three time points, TAP if this was recorded at all three time points, and mixed if there was a change from AAP to TAP at any of the three time points. For missing data, the last observation carried forward method was used to classify these subjects. As cigarette smoking is prevalent in those with SSD and has possible implications on drug metabolism, smoking status was obtained at baseline and patients were categorized as smokers or non-smokers [4,5]. Additional clinical variables, including sex and age at baseline, were also collected.

### 2.4. Genotyping, Quality Control, and Imputation of Non-Genotyped Variants 

All quality control procedures were performed using PLINK v1.9 (http://pngu.mgh.harvard.edu/purcell/plink/, accessed on 15 September 2022) according to standard protocol [31]. Complete details on genotyping and QC process were described in the Appendix A. In summary, genotype data for 2812 individuals (patients, siblings, parents, and healthy controls) and 570,038 SNVs were generated on a customized Illumina Institute of Psychological Medicine and Clinical Neurology chip array and were subjected to an extensive QC process. Non-genotyped variants were imputed using a 1000 Genome Phase 3 (version 5) reference panel on the Michigan Imputation Server (https://imputationserver.sph.umich.edu, accessed on 1 October 2022). A post imputation filter of R^2^ > 0.5 was applied to include only high quality rare variants.

### 2.5. Predicting Metabolizer Status

To generate the CYP genotype that predicted activity scores and metabolizer groups, Stargazer, a bioinformatics python tool was used [32]. We input imputed data for chromosomes 22 and 10 to identify CYP2D6 and CYP2C19 star alleles (haplotypes), respectively [32]. The tool further classified individuals into their phenotype (e.g., Intermediate metabolizers) and provided respective activity scores. To ensure accuracy, these classifications were cross-checked with the PharmVar (www.pharmvar.org, accessed on 15 January 2023) genotype–phenotype translation tables and Pharmacogenomics Knowledgebase (https://www.pharmgkb.org/, accessed on 15 February 2023) [33].

### 2.6. Polygenic Risk Scores

Unweighted PRSs were created for candidate genes, CYP2D6, and CYP2C19, as no validated (i.e., effect sizes on antipsychotics outcomes) weights were available for the SNVs on the metabolizing enzymes. To construct the unweighted PRS with SNVs associated with antipsychotic response, candidate genes, which have been studied extensively over the past 10 years, were identified through a recent meta-analysis [12]. Only SNVs that showed significant results in an additive model in Caucasians were included in the PRS. To maintain consistency with our outcome measures, these SNVs were selected based on their association with changes in symptoms. These SNVs were located on the genes HTR1A (rs6449693), HTR3A (rs1062613), COMT (rs1062613), and DRD3 (rs6280) (Appendix A). For the CYP2D6 and CYP2C19 PRSs, the core SNVs were identified through the PharmGKB and PharmVar databases (Appendix A). Due to numerous variants present in CYP2D6, unique core tier 1 SNVs identified for each star allele were selected, as sub-alleles are considered functionally equivalent, and distinguishing them is not deemed important for phenotype predictions [34]. To create unweighted PRSs for the candidate genes CYP2D6 and CYP2C19 independently, a simple scoring method was employed using the PLINK, v1.9 software (http://pngu.mgh.harvard.edu/purcell/plink/, accessed on 15 March 2023) [31]. Individuals with the presence of a risk allele were assigned a score of one, and the cumulative score represented the total number of risk alleles in their genotype. To understand the relationship between CYP2D6 and CYP2C19 PRSs, they were plotted against the CYP2D6 and CYP2C19 metabolizer phenotypes (Figure 1). Due to the relatively low number of variants in the PRSs, PRS CYP2C19 distribution was left skewed and PRS CYP2D6 was right skewed (Appendix A). Thus, all PRSs were dichotomized to high if above the median and low if below the median and incorporated as such in the models. 

The x-axis shows the unweighted PRS, which represents the number of risk alleles an individual has over the total risk alleles, and the y-axis represents the genotype–phenotype conversion based on the presence of star alleles. Abbreviation: PM, Poor metabolizer; IM, intermediate metabolizer; NM, normal metabolizer; RM, rapid metabolizer; UM, ultra-rapid metabolizer.

### 2.7. Data Analysis and Statistical Modeling 

Statistical analyses were performed using R packages version 4.1.3 (10 March 2022) [35]. The sample size for regression models was determined using the “pwr” package with a set significance level of 0.0125 (Bonferroni correction), 80% power, and moderate effect size [36]. For linear regression, a sample requirement of 202 patients was determined. For binary logistic regression, 135 patients were needed per category. For multinomial regression, 72 individuals were needed per outcome category. As complete cases were used, no missing value imputation methods were used. Logistic regression models using the “lme4” package were used to explore the relationship between unweighted PRSs and CYP phenotypic activity scores on positive and negative symptom long-term trajectories (high vs. low symptom severity) [37]. Multinomial logistic regression was used to assess the association between predictors and cognitive impairment trajectories (no, mild, high cognitive impairment) using the “nnets” package [38]. Linear regression was performed for all cardiometabolic outcomes, with all models pre-determined based on literature evidence and adjusted for age, sex, and smoking as covariates. Furthermore, three principal components were included in all models to account for the genetic population variation (refer to Appendix A). To adjust for multiple testing using the Bonferroni correction, our alpha of 0.05 was divided by the number of comparisons (four), resulting in a significant threshold of *p*-value < 0.0125. The best fitting model was selected using an adjusted R-squared or McFadden’s R-squared. A sensitivity analysis incorporating medication use (i.e., AAP, TAP, and mixed) was included in all models to determine the impact of PGx variables on clinical outcomes. For cardiometabolic symptoms that were collected at T2, medication use at this time point was considered. The best fitting model was selected using the adjusted R-squared or McFadden’s R-squared. 

## 3. Results

### 3.1. Descriptive Statistics 

In total, the study sample consisted of 691 patients (76% male) with mean age 27 ± 0.3 years, age of onset 23 ± 0.2 years, and an average illness duration of five years (Table 1). 83% were European and had on average 12 years of education. The majority of the patients were smokers (65%) and taking AAP (70%). Regarding symptom profiles, the majority of patients had low positive (71%) and negative (70%) symptom trajectories as compared to high symptom severity trajectory groups. Likewise, for cognitive symptoms, 12% had no impairments, 73% had mild impairments, whereas 15% had high cognitive impairment trajectories. Further details are presented in Table 1.

### 3.2. Metabolizer Phenotype Status 

Stargazer identified metabolizer status and activity scores for both CYP2D6 and CYP2C19 (Figure 2). CYP2D6 normal/extensive metabolizer was the most common CYP2D6 phenotype (N = 371; 54%), followed by intermediate (N = 281; 41%) and poor metabolizers (N = 39; 6%). Ten individuals with one or more haplotypes that had no function, unknown function, or uncertain function were classified as undetermined metabolizers and were excluded from further analysis, as no activity score could be generated. Ultra-rapid metabolizers were not classified, as required structure SNV (including copy number variations) were not included in genome-wide association study (GWAS) panels. However, as reported by Okhuijsen-Pfeifer and colleagues, the misclassification would have occurred in about 3% of the sample and there is no suggestive evidence that this would have a meaningful impact on the study findings [39]. For CYP2C19, individuals were classified into normal/extensive (N = 347; 50%), intermediate (N = 111; 16%), poor (N = 17; 2%), rapid (N = 182; 26%), and ultra-rapid metabolizers (N = 34; 5%). Genotype–phenotype conversion has been detailed in the Appendix A.

### 3.3. Polygenic Risk Scores 

The resulting continuous unweighted CYP2D6 PRS had a median of 0.22 (0–0.39), CYP2C19 PRS a median of 0 (0–0.50), and candidate gene PRS a median of 0.4 (0–0.80). The relationship between unweighted continuous CYP PRSs and CYP phenotypes suggests that, as more patients possess core risk alleles, their metabolism decelerates (Appendix A). Dichotomized CYP2D6 PRS based on these continuous scores classified with high score represented 60% of the sample, whereas most of the sample was classified with low CYP2C19 (62%) and candidate genes (64%) scores.

### 3.4. Long-Term Core Symptom Severity Trajectories 

Higher CYP2D6 activity scores were associated with high positive symptoms (*p* = 0.01) and high cognitive impairments (*p* = 0.0005) as compared to low symptom severity groups (Figure 3). Individuals with high CYP2D6 activity scores had 1.46 and 2.53 greater odds of experiencing positive symptoms and cognitive impairments, respectively, as compared to those without impairments. Notably, smoking status as a covariate was also significantly associated with high positive symptoms (*p* = 0.001) and both mild (*p* = 0.009) and high (*p* = 0.001) cognitive impairment groups. This suggests that smokers have 1.90 greater odds of experiencing high positive symptoms, 1.92 greater odds of mild cognitive impairments, and 2.97 greater odds of high cognitive impairment compared to low symptom severity groups. No associations were found between predictors and negative symptom severity groups. No significant associations were found between candidate gene PRS, CYP2D6 PRS, and CYP2C19 PRS and symptom domains (Figure 3; Appendix A). In the sensitivity analysis accounting for medication use, those who switched medications (mixed use) had 1.81 greater odds of experiencing high negative symptoms compared to those with continued AAP use; however, this was nominally significant (*p* = 0.04). No significant differences were found between those using TAP and AAP with positive and cognitive symptom severity groups. 

### 3.5. Cardiometabolic Variables 

No PRSs or CYP activity scores were associated with any cardiometabolic outcome (Appendix A). Smoking status was nominally associated with pulse rate (*p* = 0.03), indicating that smokers have an increase in pulse rate of 3.5 beats/min relative to non-smokers. In the sensitivity analysis accounting for medication use, AAP was significantly associated with pulse rate (*p* = 0.003), indicating an increased pulse rate by 8.15 beats/min for each AAP user relative to those taking TAP. Additionally, when accounting for medication use, those classified with a high CYP2D6 PRS score (slow CYP2D6 metabolism) were found to have an increased BMI of 1.2 kg/m^2^ relative to those with low CYP2D6 PRS scores (fast CYP2D6 metabolism); however, this was only nominally significant (*p* = 0.02).

## 4. Discussion

The present study investigated the influence of unweighted PGx PRS in antipsychotic treatment on long term clinical trajectories and cardiometabolic complications in patients with SSD. Drawing on existing literature and guidelines, the study utilized genotype data from 691 patients with European ancestry to generate unweighted PRS for CYP2D6, CYP2C19, and candidate genes as well as CYP2D6 and CYP2C19 activity scores. While these PRSs did not significantly associate with clinical and cardiometabolic outcomes, higher CYP2D6 activity scores were found to play a role in high positive symptom and high cognitive impairment trajectories as compared to low symptom severity groups. Additionally, we observed that the covariate smoking was significantly associated with positive and cognitive symptom severity, but not with negative symptoms. These findings have important implications for the future directions of personalized medicine in psychiatry. 

The activity score of CYP2D6 was significantly associated with long term positive and cognitive symptom trajectories, indicating that those with higher CYP2D6 functional activity are likely to have higher symptom severity. This is supported by the notion that reduced drug plasma concentrations decrease the effectiveness of the medication [40]. Studies investigating the role of CYP in patients on clozapine have found opposite results indicating no association between CYP functional activity and change in PANSS overall symptom severity scores [30,41]. Furthermore, a recent systematic review demonstrated mixed findings regarding CYP2D6’s role, with most studies reporting no significant association with PANSS overall scores and few reporting slight associations with subscales of positive domains [42]. This suggests that CYP2D6, among others, may play a role in medications that target positive and cognitive symptom domains, however, more research is needed to uncover its true association. 

Although cognitive impairments have not been extensively studied in the PGx of antipsychotics, there has been growing interest due to their high heritability and severe impact on quality of life [29]. Our study contributes to the emerging critical role of CYP2D6 in cognition, which is widely expressed in the brain [43]. Core SNV rs16947 of CYP2D6 has been associated with both cognition and schizophrenia in previous research [44,45,46]. Studies have found that increased CYP2D6 metabolism is linked with reduced cognitive function, particularly in attentional processing [43]. Similarly, another study found that carriers of the CYP2D6*4 allele (NM) were associated with cognitive impairments, suggesting CYP2D6 genotype may moderate cognition [47]. 

Our results did not associate CYP2C19 with core symptom domains, but previous studies have found mixed findings for overall PANSS symptom severity with certain antipsychotics [39,41]. Lesche et al. found no association between CYP2C19 PM and lower symptom severity, while another study found a significant association between higher CYP2C19 activity scores and lower symptoms for clozapine users [39,41]. These mixed results may explain why there are no guidelines with recommendations for CYP2C19 metabolizers and antipsychotics yet [20]. 

Regarding cardiometabolic parameters, no association was found between CYP2D6 and CYP2C19 on metabolic parameters, consistent with previous studies [21,30,48]. While most studies have failed to find an association between different CYP2D6 metabolizing activity and cardiometabolic parameters, they did not differentiate between the effects of specific antipsychotics, often due to small sample sizes [30,48]. A comprehensive meta-analysis of antipsychotics and cardiometabolic parameters revealed that different antipsychotics have varying effects, with clozapine and olanzapine accounting for the most metabolic dysregulation [49]. Thus, it is essential to analyze the efficacy of antipsychotics separately [49]. 

A strong association between smoking and more severe positive and cognitive symptom domains was found. Although some studies have adjusted for smoking status to avoid phenoconversion, others have concluded that smoking alone explains more variance in symptom severity than metabolizing status [41]. While the relationship between smoking and CYP1A2 is well established, that is not the case for CYP2D6 and CYP2C19 [50]. Despite not accounting for smoking in the CYP activity scores, our results align with the larger body of literature indicating that the role of smoking behavior is significantly associated with increased positive symptoms and cognitive impairment [51,52,53,54]. While 70–80% of those with SSD are smokers, nicotine has been associated with reduced negative and extrapyramidal side effects through its activation of the nicotinic acetylcholine receptors, leading to its use as “self-medication” by many patients [51,55,56].

Lastly, we found that combining previously studied candidate gene variants in a PRS did not improve predictability, likely due to the lower power of biologically derived PRS with few mechanistic SNVs. Notably, none of these SNVs have been identified in recent hypothesis-free GWAS of antipsychotic response [57,58]. While the dopamine dysfunction hypothesis has been a focus in the past decade, these results suggest that other receptor systems might be more important, including the role of glutamate receptor systems, particularly genes GRID2, GRM7, SLC1A1, and TNIK [58]. Glutamate receptors have also recently been linked to the etiology of schizophrenia, suggesting its abnormal functioning in many regions of the brain may contribute to its development [59]. Thus, GWAS may be essential to better understand the mechanism of antipsychotics and identify novel drug targets for precision PGx treatment [6,7]. Furthermore, they should focus on analyzing rare variants to capture small variants with large effect sizes, such as that of the CYP enzymes. While the current study did not find CYPD2D6 PRS and CYC19 PRS based on core variants to be significant predictors of antipsychotic response, it reveals the potential importance of star alleles, which can only be calculated through the identification of several SNVs [32]. As there have been emerging studies finding significant associations with star alleles rather than the genotype predicted metabolizer derived from PGx guidelines, they should continue to be investigated [40,47]. Considering there are numerous combinations of haplotypes that can result in normal or intermediate metabolizer phenotypes, we may fail to identify a critical role of certain star alleles. Evaluating unknown function alleles individually may help determine their function. 

The current study offers several noteworthy strengths and limitations. By parsing the heterogeneity in symptom severity through investigating longitudinal symptom trajectories, we were able to accurately assess the role of PGx across positive, negative symptom domains, and cognitive functioning. Our findings emphasize the need for a personalized approach toward treatment plans that consider the distinct pathophysiology of each symptom domain rather than overall symptom severity [8,9]. Moreover, our findings suggest that a hypothesis-free GWAS approach may advance the field [10,12]. This aligns with the recommendations of Siemens et al. (2022), who conducted a systematic review of PGx PRS and found large discrepancies between those of candidate gene studies and GWAS, suggesting future studies should be hypothesis-free [60]. Lastly, as drug metabolism is known to differ by ancestry groups, this study utilized a strict sample of those with European ancestry only, which may increase clinical utility [50,61]. However, we lacked the statistical power to compare individual antipsychotics, which may account for the lack of effect seen in the sensitivity analysis on antipsychotic use as well as on cardiometabolic outcomes. We also lacked information on drug blood concentration levels, which is key in understanding the dose–metabolism relationship, although previous studies have not found significant relationships [39,41]. Additionally, as individuals with SSD often have polypharmacy and comorbidities, our study’s failure to consider these factors represents a key limitation [41,50,62]. Finally, the GROUP cohort is a relatively healthy sample, which may explain the lack of effect found between CYP and cardiometabolic symptoms.

## 5. Conclusions

No association was found between pharmacogenetic PRSs, apart from CYP2D6 function, when adjusting for smoking status. We found that increased CYP2D6 activity score was associated with increased longitudinal positive and cognitive symptom severity trajectories, but not with negative symptom trajectories or cardiometabolic outcomes. These findings highlight the potential clinical value of pre-emptive genetic testing for the functional activity of this metabolizing enzyme. This assessment considers an individual’s genetic variations in this enzyme which can then categorize them as poor (PM) or rapid (RM), for example. This assessment can thus possibly aid in treatment selection for the most appropriate antipsychotics and dose, which can lead to a better treatment response. Additionally, our study highlights the potential need for clinicians to evaluate the improvement of symptoms through specific domains in carefully designed naturalistic studies to better understand how antipsychotics may be influencing their treatment. 

By tailoring antipsychotic choices and dosages based on an individual’s functional CYP enzyme activity, physicians can enhance desired outcomes while minimizing the occurrence of side effects and adverse reactions. This personalized approach to treatment can lead to more effective and safer antipsychotic therapy in psychiatry.

## Figures and Tables

**Figure 1 jpm-13-01354-f001:**
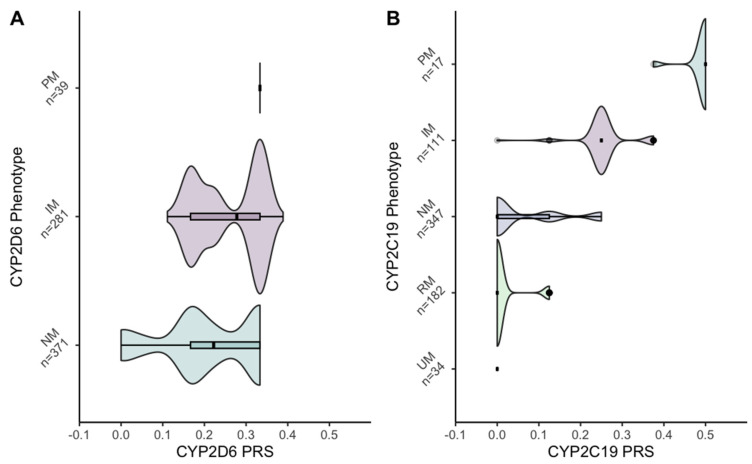
Violin plot showing the relationship between CYP2D6 PRS and CYP2D6 phenotype (**A**) and CYP2C19 PRS with CYP2C19 phenotype (**B**).

**Figure 2 jpm-13-01354-f002:**
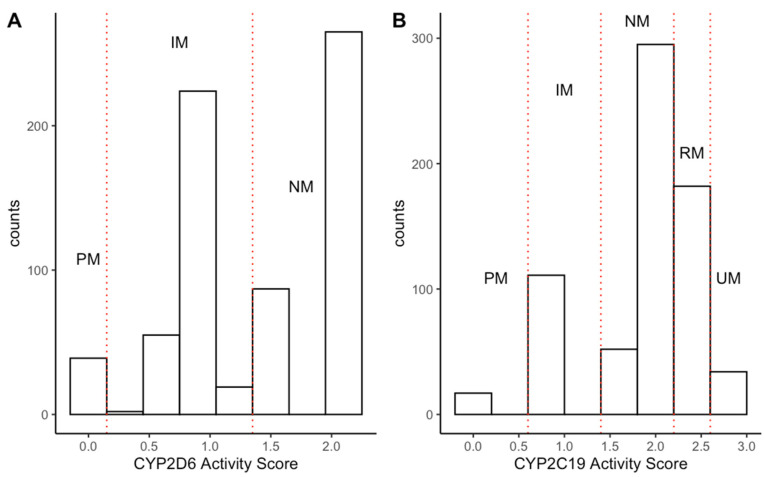
Distribution of CYP2D6 (**A**) and CYP2C19 (**B**) phenotypes based on genotype–phenotype conversion. Red dashed lines indicate the separation of metabolizer-phenotype classifications. The x-axis shows the CYP activity scores and the y-axis reflects the total number of individuals classified as each metabolizer phenotype. Red dashed lines indicate the separation of metabolizer-phenotype classifications. Abbreviation: PM, Poor metabolizer; IM, intermediate metabolizer; NM, normal metabolizer; RM, rapid metabolizer; UM, ultra-rapid metabolizer.

**Figure 3 jpm-13-01354-f003:**
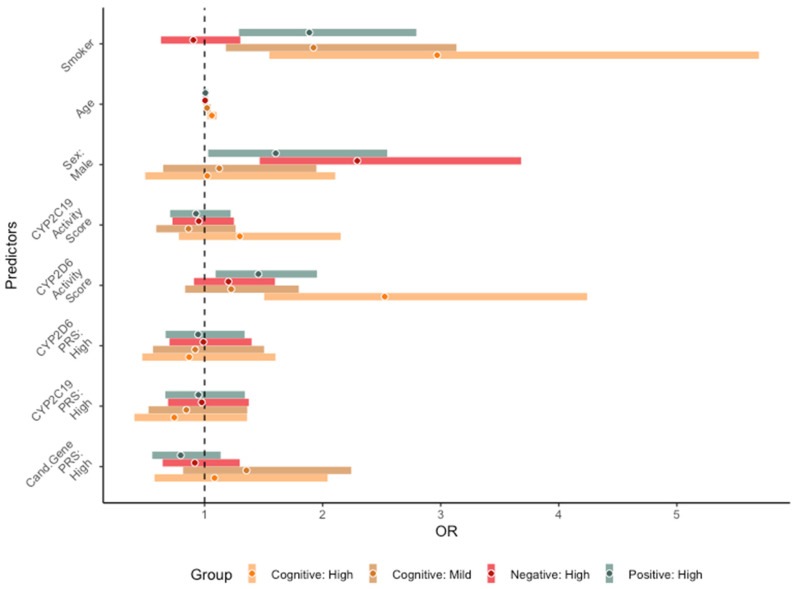
Forest plot for core symptom trajectories. Odds ratios and respective 95% confidence intervals are plotted for each outcome. The x-axis reflects the odds ratio and the y-axis shows the predictor variables. Groups with high symptom severity are plotted relative to low symptom severity groups for negative and positive symptoms. Highly impaired and mildly impaired cognitive function groups are plotted relative to the no impairment group. Unweighted dichotomized PRSs (CYP2D6 PRS, CYP2C19 PRS, candidate gene PRS) have been adapted from a separate model (to avoid collinearity). High PRS groups are compared relative to low PRS groups.

**Table 1 jpm-13-01354-t001:** Demographic and clinical characteristics of the 691 studied patients.

Baseline Measures		Longitudinal Measures (6 Year Follow-Up)	
Age (years), mean (SE)	27.71 (0.27)	Antipsychotic use N (%):	
Sex (male), N (%)	525 (75.98)	AAP use	485 (70.19)
Education (years), mean (SE)	12.46 (0.15)	TAP use	42 (6.08)
Smokers N (%)	449 (64.98)	Mixed (AAP & TAP)	57 (8.25)
Age onset illness (years), mean (SE)	23.19 (0.23)	Unknown	107 (15.48)
Duration of Illness (years), mean (SE)	4.97 (0.17)	Positive symptoms N (%):	
Psychotic episodes, mean (SE)	1.72 (1.17)	Low	491 (71.06)
Systolic BP (mmHg), mean (SD)	127.35 (15.68)	High	200 (28.94)
Diastolic BP (mmHg), mean (SD)	79.38 (11.36)	Negative symptoms N (%):	
Pulse rate (beat/min), mean (SD)	75.30 (15.80)	Low	489 (70.77)
Triglycerides (mmol/L), median (IQR)	1.84 (1.30)	High	202 (29.23)
HDL (mmol/L), mean (SD)	1.27 (0.74)	Cognitive impairments N (%):	
LDL (mmol/L), mean (SD)	3.12 (0.94)	None	84 (12.16)
HbA1c (mmol/mol), mean (SD)	35.01 (6.05)	Mild	500 (72.36)
BMI (kg/m^2^), mean (SD)	26.01 (4.72)	High	107 (15.48)
Waist circumference (cm), mean (SD)	95.08 (14.13)		

Abbreviations: SE, standard error; BP, blood pressure; HDL, high-density-lipoprotein; LDL, low-density-lipoprotein; HbA1c, glycated hemoglobin; BMI, body mass index; AAP, atypical antipsychotic; TAP, typical antipsychotic.

## Data Availability

Data pertain to the Genetic Risk and Outcome of Psychosis (GROUP) project. Data are available upon request from researchers who meet the criteria for data access. Researchers can send their request to the coordinators of the GROUP study by sending an email to j.vanbaaren@umcutrecht.nl.

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
