# Peer review of "Pharmacogenetics of Long-Term Outcomes of Schizophrenia Spectrum Disorders: The Functional Role of CYP2D6 and CYP2C19"

_jpm, 2023, doi:10.3390/jpm13091354_

Round 1

Reviewer 1 Report

The following study successfully introduces the complex nature of SSD, the challenges of antipsychotic treatment, and the relevance of genetic variation in treatment response. The rationale for the current study is convincingly presented, considering the limitations of previous PGx studies and the need for larger-scale, comprehensive investigations. The discussion effectively synthesizes the study's findings, offering a clear interpretation of the observed associations and their relevance to personalized medicine in psychiatry. The authors appropriately acknowledge limitations while pointing to potential future research directions. Overall, this study adds depth and context to the findings and their implications.

However, I have few minor suggestions:

The introduction touches upon the focus of the study on "unweighted PGx PRS," but it would be beneficial to briefly explain the rationale behind using unweighted PRS. This would help readers grasp why this approach was chosen and its potential advantages.

The methods section is detailed and well-structured, the explanations for some terms, such as "Stargazer" for haplotype identification, could be briefly expanded upon to ensure that readers not familiar with these tools can follow the methods with ease.

In the discussion of CYP2D6's association with symptom domains, it might be beneficial to discuss the potential clinical implications of these findings. For instance, how might CYP2D6 activity inform treatment strategies for patients with different symptom severities or trajectories? This could enhance the practical implications of the study's results.

The discussion mentions the emerging role of glutamate receptor systems in antipsychotic response and the potential importance of GWAS to uncover novel drug targets. To strengthen this point, the authors could provide a brief overview of the significance of glutamate receptor systems in the context of SSD and antipsychotic treatment.

Author Response

We like to thank this reviewer for his/her positive evaluation of helpful comments and suggestions for our manuscript and for providing us with valuable feedback. Here below we address the comments per points.

Comments and Suggestions for Authors

The following study successfully introduces the complex nature of SSD, the challenges of antipsychotic treatment, and the relevance of genetic variation in treatment response. The rationale for the current study is convincingly presented, considering the limitations of previous PGx studies and the need for larger-scale, comprehensive investigations. The discussion effectively synthesizes the study's findings, offering a clear interpretation of the observed associations and their relevance to personalized medicine in psychiatry. The authors appropriately acknowledge limitations while pointing to potential future research directions. Overall, this study adds depth and context to the findings and their implications.

Reply: we thank the reviewer for this concise affirmation of the appropriateness of our study.

However, I have few minor suggestions:

Comment 1: The introduction touches upon the focus of the study on "unweighted PGx PRS," but it would be beneficial to briefly explain the rationale behind using unweighted PRS. This would help readers grasp why this approach was chosen and its potential advantages.

Reply: thank you for notifying this issue. We added the rationale behind using unweighted PGX PRS in method section on page 4 lines 211 to 213 as the following:

“Unweighted PRSs were created for candidate genes, CYP2D6, and CYP2C19 as no validated weights (i.e. effect sizes on antipsychotics outcomes) were available for the SNVs on the metabolizing enzymes.”

Comment 2: The methods section is detailed and well-structured, the explanations for some terms, such as "Stargazer" for haplotype identification, could be briefly expanded upon to ensure that readers not familiar with these tools can follow the methods with ease.

Reply: Thank you for this suggestion. We clarify further the “Stargazer" for haplotype identification” in the method section on pages 4 lines 203 to 205 as the following:

“To generate the CYP genotype predicted activity scores and metabolizer groups, Stargazer, a bioinformatics python tool was used[32]. We input imputed data for chromosome 22 and 10 to identify CYP2D6 and CYP2C19 star alleles (haplotypes) respectively[32].”

Comment 3: In the discussion of CYP2D6's association with symptom domains, it might be beneficial to discuss the potential clinical implications of these findings. For instance, how might CYP2D6 activity inform treatment strategies for patients with different symptom severities or trajectories? This could enhance the practical implications of the study's results.

Reply: Thank you for this very useful suggestion. We added the following statement in the conclusion section, page 10, lines 499 to 529:

These findings highlight the potential clinical value of pre-emptive genetic testing for the functional activity of this metabolizing enzyme. This assessment considers an individual’s genetic variations in this enzyme which can then categorize them as poor (PM or rapid (RM) for example. This assessment can thus possibly aid in treatment selection for the most appropriate antipsychotics and dose which can lead to a better treatment response. Additionally, our study highlights the potential need for clinicians to evaluate the improvement of symptoms through specific domains in carefully designed naturalistic studies to better understand how antipsychotics may be influencing their treatment.

By tailoring antipsychotic choices and dosages based on an individual's functional CYP enzyme activity, physicians can enhance desired outcomes while minimizing the occurrence of side effects and adverse reactions. This personalized approach to treatment can lead to more effective and safer antipsychotic therapy in psychiatry.

Comment 4: The discussion mentions the emerging role of glutamate receptor systems in antipsychotic response and the potential importance of GWAS to uncover novel drug targets. To strengthen this point, the authors could provide a brief overview of the significance of glutamate receptor systems in the context of SSD and antipsychotic treatment.

Reply: We agree with the reviewer. To further elaborate on the possible role of the glutamate receptor in the context of SSD we added the following statement on page 10 , lines 461 to 462 .

“Glutamate receptors have also recently been linked to the etiology of schizophrenia, suggesting its abnormal functioning in many regions of the brain may contribute to its development [59].”

Reviewer 2 Report

The manuscript by Sandhu et al. is a well-conducted study that provides some interesting insights into the role of pharmacogenetics in schizophrenia. The authors' findings suggest that PRS are not useful for predicting clinical outcomes in schizophrenia, but that CYP2D6 activity scores may be a useful biomarker for predicting cognitive impairment. I would be interested to see if future studies replicate these findings and explore the potential clinical utility of CYP2D6 activity scores in schizophrenia.

Some typos are still there, but the English language is fine. 

Author Response

to hear that you found it to be well-conducted and that it offers some interesting insights.

Comments and Suggestions for Authors

Comment 1: The manuscript by Sandhu et al. is a well-conducted study that provides some interesting insights into the role of pharmacogenetics in schizophrenia. The authors' findings suggest that PRS are not useful for predicting clinical outcomes in schizophrenia, but that CYP2D6 activity scores may be a useful biomarker for predicting cognitive impairment. I would be interested to see if future studies replicate these findings and explore the potential clinical utility of CYP2D6 activity scores in schizophrenia.

Reply: Thank you for this interesting comment. We added the following statement in the discussion, on page 10, lines 497 to 527explaining the potentials of CYP activity score as a biomarker in predicting the antipsychotics outcomes in SSD:

“These findings highlight the potential clinical value of pre-emptive genetic testing for the functional activity of this metabolizing enzyme. This assessment considers an individual’s genetic variations in this enzyme which can then categorize them as poor (PM or rapid (RM) for example. This assessment can thus possibly aid in treatment selection for the most appropriate antipsychotics and dose which can lead to a better treatment response. Additionally, our study highlights the potential need for clinicians to evaluate the improvement of symptoms through specific domains in carefully designed naturalistic studies to better understand how antipsychotics may be influencing their treatment.

By tailoring antipsychotic choices and dosages based on an individual's functional CYP enzyme activity, physicians can enhance desired outcomes while minimizing the occurrence of side effects and adverse reactions. This personalized approach to treatment can lead to more effective and safer antipsychotic therapy in psychiatry.”

Comment #2: Comments on the Quality of English Language: Some typos are still there, but the English language is fine. 

Reply: thank you for notifying these typos. We have edited the manuscript carefully and corrected any overlooked typos though out the text and supplementary.

Reviewer 3 Report

Title: maybe consider making it informative

Abstract: no balanced – 60% of the amount describing aim with no details of methods.

Keywords: consider genotyping  and  Cytochrome P450 2D6

use a phrase that introduction need to describe the scientific context and justification for the reported study. I will suggest changing Schizophrenia Spectrum Disorders (SSD) with ICD/DSM language.

Lines 36-44 need shortening.

Line 76-89 clarify: 1. How the presence of rare functional SNVs in CYP2D6 and other relevant enzymes may impact the metabolism of antipsychotic medications?

I suggest to add graph/figure Poor metabolizers (PM) vs intermediate (IM), normal/extensive (NM), or rapid/ultra-rapid (RM/UM) metabolizing phenotypes and adverse drug reactions or toxicities especially anti-psychotics.

Lines 89-95 very repetitive writing style.

describe the scientific context (i.e., GROUP) and justification for the reported study. I don’t this its justified to say: “Further details on the structure of the GROUP cohort can be found in a previous publication[23]” I suggest to make it clear for readers in one paper all details.

describe the context, the places, the pertinent times, such as the recruiting, exposure, follow-up, and data collection times.

describe your pre-established hypothesis.

give the qualifying requirements for incl vs excl, as well as the sources and procedures used to choose the participants. Who performed diagnosis(team member of the research authors or independent) and is it Diagnostic and Statistical Manual of Mental Disorders (DSM-IV). I thought Italian national regulations formally require the ICD labels.

Describe equipment’s or validation for all outcomes, exposures, predictors, potential confounders, and effect modifiers should be precisely defined. E.g. Cronbach alpha or Italian PANSS.

describe how the study size was determined.

describe the methods used to handle quantitative variables in the analyses. describe the classifications that were chosen, if relevant, and why.

describe all statistical techniques, including confounding correction techniques.

describe how missing data were handled.

Results are very clear.

discuss how broadly (externally valid) the study's findings can be applied to clinicians.

Author Response

We thank this reviewer for detailed comments on our manuscript and for providing us with valuable feedback.

Comments and Suggestions for Authors 

Comment 1:Title: maybe consider making it informative 

Reply: : To make the title more clear we have made the following adjustment on page 1 lines 1 to 4 as the following:

“Pharmacogenetics of long-term outcomes of Schizophrenia Spectrum Disorders: The functional role of CYP2D6 and CYP2C19”

Comment 2: Abstract: no balanced – 60% of the amount describing aim with no details of methods. 

Reply: Thank you for highlighting this issue. We have added an additional statement on the methods section for more clarity on page 1 line 25-27:

“Furthermore, CYP2D6 and CYP2C19 functional activity scores were calculated to determine the relationship between metabolism and clinical outcomes.”

Comment 3 Keywords: consider genotyping  and  Cytochrome P450 2D6. use a phrase that introduction need to describe the scientific context and justification for the reported study. I will suggest changing Schizophrenia Spectrum Disorders (SSD) with ICD/DSM language.

Reply: Thank you for the suggestion to include these key words. We have added them in the key words section for on page 1 line 34 as:

“genotyping; Cytochrome P450.”

Comment 4 Lines 36-44 need shortening. 

Reply: Thank you for encouraging us to be more concise. As per your suggestion we have shortened the introduction on page 1 lines 37 to 42 as:

“Schizophrenia Spectrum Disorders (SSD) are severe mental disorders that affect around 1% of the global population[1]. Various combinations of positive symptoms (i.e., hallucinations), negative symptoms (i.e., social withdrawal), and cognitive impairments (e.g., memory problems) contribute to its impairments on quality of life and mortality[1,2]. Despite antipsychotic medication being mainstay treatment, their effectiveness is highly variable; up to 50% of patients do not experience symptom relief or experience adverse drug effects[3–5].”

Comment 5 Line 76-89 clarify: 1. How the presence of rare functional SNVs in CYP2D6 and other relevant enzymes may impact the metabolism of antipsychotic medications? 

Reply: Good point to clarify these lines. We have made a minor change in language to clarify the role of CYP2D6 metabolism may be important in antipsychotic medication on page 2 lines 102-103 as:

“Individuals with CYP2D6 PM, which occur in 10% of Europeans, are thought to be at significantly at higher risk for adverse drug events and lack of response for antipsychotics compared to other metabolizers[12].”

Comment 6 I suggest to add graph/figure Poor metabolizers (PM) vs intermediate (IM), normal/extensive (NM), or rapid/ultra-rapid (RM/UM) metabolizing phenotypes and adverse drug reactions or toxicities especially anti-psychotics. 

Reply: Thank you for this comment- it would indeed be helpful to have a visualization on how these metabolizing phenotypes relate to adverse drug reactions. However, the evidence presented to date is not clear and the regulatory bodies have poor agreement for which metabolism phenotypes are important. Since the guidelines and data are not clear, we cannot make a graph to which metaboliser phenotypes have greater risk to antipsychotics as this is the need to further investigate.

Comment 7 Lines 89-95 very repetitive writing style. 

Reply:  Thank you for this comment. We have been repetitive to really drive home the point on the lack of consistency with evidence in research and from regulatory bodies for which metabolizing enzymes play a key role in antipsychotic metabolism and which phenotypes of these are deemed most important.

Comment 8 describe the scientific context (i.e., GROUP) and justification for the reported study. I don’t this its justified to say: “Further details on the structure of the GROUP cohort can be found in a previous publication[23]” I suggest to make it clear for readers in one paper all details.

describe the context, the places, the pertinent times, such as the recruiting, exposure, follow-up, and data collection times.

Reply: This is a good suggestion. We have added more details of GROUP so readers have better insight into what the study entailed. This is reflected in the methods section on page 3 lines 128-140 as:

“The present study is situated within the framework of the Genetic Risk and Outcome of Psychosis (GROUP) cohort[23]. GROUP is a comprehensive, six-year, prospective national study in the Netherlands, which aimed to investigate the role of genetic and environmental factors which contribute to the expression of psychosis. GROUP involved 36 partner mental institutions and university medical centers (Amsterdam, Groningen, Utrecht, Maastricht). Clinicians working within these institutions screened patients for eligibility based on the following inclusion criteria: (1) have schizophrenia spectrum disorder according to the 4th edition of the Diagnostic and Statistical Manual of Mental Disorders (DSM-IV), (2) be between the ages of 16-50, (3) be Dutch speakers and, (4) be able to provide informed consent[2]. In addition to patients, unaffected first-degree relatives of patients and unrelated healthy controls were also sampled. Clinical outcome data was collected at one of the respective institutions or medical centers at baseline(T1), three (T2), and six (T3) years along with genotype data.”

Comment 9 describe your pre-established hypothesis9

Reply: Good suggestion. We have added the pre established hypothesis on page 2 lines 117 to 125 as:

“We hypothesized, combining significant SNVs identified from a meta-analysis of candidate gene studies into a PRS would increase predictive power and result in a significant association with clinical outcomes. Furthermore, we proposed PRS for CYP2D6 and CYP2C19 enzymes would be effective in determining an individual’s metabolizer status and be associated with clinical outcomes. This study took on a polygenic approach to determine the involvement of key PGx variants on long-term clinical symptom trajectories and cardiometabolic parameters.”

Comment 10 give the qualifying requirements for incl vs excl, as well as the sources and procedures used to choose the participants. Who performed diagnosis(team member of the research authors or independent) and is it Diagnostic and Statistical Manual of Mental Disorders (DSM-IV). I thought Italian national regulations formally require the ICD labels.

Reply: Thank you for this suggestion we have added this on page 3 lines 133-140 as:

“Clinicians working within these institutions screened patients for eligibility based on the following inclusion criteria: (1) have schizophrenia spectrum disorder according to the 4th edition of the Diagnostic and Statistical Manual of Mental Disorders (DSM-IV), (2) be between the ages of 16-50, (3) be Dutch speakers and, (4) be able to provide informed consent[2]. In addition to patients, unaffected first-degree rel-atives of patients and unrelated healthy controls were also sampled. Clinical outcome data was collected at one of the respective institutions or medical centers at baseline(T1), three (T2), and six (T3) years along with genotype data.”

Comment 11 Describe equipment’s or validation for all outcomes, exposures, predictors, potential confounders, and effect modifiers should be precisely defined. E.g. Cronbach alpha or Italian PANSS.

Reply: Thank you for this suggestion. As every tool used has been referenced from its original publication we believe there is sufficient information. However, a bit more detail but not to lose the reader in this section a publication on the psychometrics of the PANSS has been included on page 3 lines 143-145 as:

“To assess the severity of positive and negative symptoms in patients, the widely used valid and reliable Positive and Negative Syndrome Scale (PANSS) for Schizophrenia was employed[24].”

Comment 12 describe how the study size was determined.

Reply: Thank you for requesting more details on this. We have included this information on page 5 lines 247-251 as:

“Sample size for regression models was determined using the “pwr” package with a set significance level of 0.0125 (Bonferroni correction), 80% power, and moderate effect size. For linear regression a sample requirement of 202 patients was determined.  For binary logistic regression, 135 patients were needed per category. For multinomial regression, 72 individuals were needed per outcome category.”

Comment 13 describe the methods used to handle quantitative variables in the analyses. describe the classifications that were chosen, if relevant, and why. 

Reply: Thank you for this suggestion, we have added the quantitative cut off for our main outcome variables on page 3 lines 159-162 as:

Positive and negative symptom trajectories were dichotomized into high and low symptom groups, while cognition impairments were consolidated into high, mild, and no impairment groups. This resulted in sum scores <15 for both positive and negative symptoms classifying as low severity and >15 were for high symptom severity. For the cognitive composite scores, high impairments were <-2, mild were 1 to -1, and no impairments were > 1.5.

Comment 14 describe all statistical techniques, including confounding correction techniques.

Reply: To further elaborate on our statistical methods we have included how we selected the models in the methods section page 5 lines 263 to 264 as:

“The best fitting model was selected using adjusted R-squared or McFadden's R-squared.”

Comment 15 describe how missing data were handled.

Reply: As per request we have clarified how missing data was handled in the methods section on page 5 lines 251 to 252 as:

“As complete cases were used, no missing value imputation methods were used.”

Comment 16 Results are very clear. 

Reply: Thank you for acknowledging our efforts to be clear in the results.

Comment 17 discuss how broadly (externally valid) the study's findings can be applied to clinician

Reply:  Thank you for this suggestion for a broader approach. We added the following statement in the discussion, on page 10 line 499 to 529:

“These findings highlight the potential clinical value of pre-emptive genetic testing for the functional activity of this metabolizing enzyme. This assessment considers an individual’s genetic variations in this enzyme which can then categorize them as poor (PM or rapid (RM) for example. This assessment can thus possibly aid in treatment selection for the most appropriate antipsychotics and dose which can lead to a better treatment response. Additionally, our study highlights the potential need for clinicians to evaluate the improvement of symptoms through specific domains in carefully designed naturalistic studies to better understand how antipsychotics may be influencing their treatment.

By tailoring antipsychotic choices and dosages based on an individual's functional CYP enzyme activity, physicians can enhance desired outcomes while minimizing the occurrence of side effects and adverse reactions. This personalized approach to treatment can lead to more effective and safer antipsychotic therapy in psychiatry.”

Round 2

Reviewer 3 Report

thank you for addressing my concerns.

English language is fine. I still feel some repititive tone but authors are aware of it and intentionally want to keep it. it's not affecting the science of the paper.